# Factors to Male-Female Sex Approaches and the Identification of Volatiles and Compounds from the Terminalia of *Proholopterus chilensis* (Blanchard) (Coleoptera: Cerambycidae) Females in *Nothofagus obliqua* (Mirb.) Oerst. (Nothofagaceae) Forests in Chile

**DOI:** 10.3390/insects15100741

**Published:** 2024-09-26

**Authors:** Diego Arraztio, Amanda Huerta, Andrés Quiroz, Washington Aniñir, Ramón Rebolledo, Tomislav Curkovic

**Affiliations:** 1Doctorado en Ciencias Silvoagropecuarias y Veterinarias, Facultad de Cs. Agronómicas, Universidad de Chile, Santiago P.O. Box 1004, Chile; dsarrazt@uchile.cl; 2Facultad de Ciencias Forestales y de la Conservación de la Naturaleza, Universidad de Chile, Santiago P.O. Box 9206, Chile; ahuerta@uchile.cl; 3Departamento de Ciencias Químicas y Recursos Naturales, Universidad de La Frontera, Temuco 4811230, Chile; andres.quiroz@ufrontera.cl; 4Centro de Investigación Biotecnológica Aplicada al Medio Ambiente (CIBAMA), Universidad de La Frontera, Temuco 4811230, Chile; 5Doctorado en Ciencias de Recursos Naturales, Universidad de La Frontera, Temuco 4811230, Chile; w.aninir01@ufromail.cl; 6Facultad de Ciencias Agropecuarias y Medio Ambiente, Universidad de La Frontera, Temuco 4811230, Chile; ramon.rebolledo@ufrontera.cl; 7Facultad de Ciencias Agronómicas, Universidad de Chile, Santiago 8820808, Chile

**Keywords:** Cerambycinae female-produced sex pheromone, Cerambycidae sex ratio, Cerambycinae trail-pheromones

## Abstract

**Simple Summary:**

*Proholopterus chilensis* (Cerambycidae: Cerambycinae) is a native species and an important damage agent in the Nothofagaceae forests of southern Chile, significantly affecting timber production in some areas, especially production of *Nothofagus obliqua*, with no sanitary management measures currently available. Therefore, the present research determined the predisposing factors for male–female sex approaches and identified, for the first time, some compounds that appear to mediate sexual behaviors that could be used to develop specific, safe, and effective control strategies. Interestingly, the chemical communication for mating purposes in *P. chilensis* preliminarily seems to include both possible “volatile or airborne” pheromones (eventually long-range ones) and short-range pheromonal compounds produced only by females, something, as far as we know, unique in Cerambycinae.

**Abstract:**

During the spring–summer seasons between 2019 and 2023, in the localities of Maquehue (La Araucanía Region) and Llifén (Los Ríos Region), we collected 262 virgin *Proholopterus chilensis* (1 female/2.3 males) specimens emerging from the live trunks of *N. obliqua* trees, an atypical sex ratio in Cerambycidae, suggesting high male competition for females. Virgin specimens of both sexes were individually placed in panel traps in the field, capturing only males (n = 184) over the field study seasons and only in traps baited with females. This fact preliminarily suggests the *P. chilensis* females emit possible “volatile or airborne” pheromones (eventually being a long-range sex pheromone), something unusual in Cerambycinae, the subfamily to which it currently belongs, although the taxonomic status of the species is under debate. In Llifén and Santiago (Metropolitan Region), behavioral observations were conducted, which allowed us to define the conditions necessary for male–female encounters that were replicated when carrying out volatile captures (Head Space Dynamic = HSD) and collections of compounds from terminalias excised from females. In field trials, virgin females less than ten days old were more attractive than older ones and attracted males during the night, i.e., between 23:00 and 5:00 AM, when the ambient temperature exceeded 11.6 °C. The aeration of females under the conditions described above and subsequent analysis of extracts by GC-MS allowed the identification of compounds absent in males and the control, including two oxygenated sesquiterpenes, a nitrogenous compound (C_20_), and a long-chain hydrocarbon (C_26_). From the terminalia extracts, hentriacontane, heptacosane, and heneicosyl, heptacosyl and docosyl acetates were identified by GC-MS, and their roles are unknown in the development of short-distance sexual behaviors, but they could serve to mark a trail leading the male towards the female in the final stage of approach for courtship/mating. Thus, we proposed the hypothesis that *P. chilensis* females emit both a long-range and a trail-pheromone, which, if confirmed, would be a rare case in this family. The specific identity of the compounds obtained by HSD, as well as the activities of these chemicals and those obtained from the terminalias, should be evaluated in future behavioral studies, as well as regarding their potential to attract males under field conditions. The current document is the first report on volatiles obtained from aerations and compounds extracted from female terminalia in *P. chilensis*.

## 1. Introduction

*Proholopterus chilensis* (Blanchard) (Cerambycidae) is a long-horned beetle currently placed in the subfamily Cerambycinae, tribe Proholopterini, which includes species present only in Chile and Argentina [1]. In Chile, it is present between the regions of Maule (−35.4333, −71.6666) and Los Lagos (−41.4716, −72.9366) [2,3]. Its larval stage develops only in live trees of *Nothofagus obliqua* (Mirb.) Oerst., *Nothofagus alpina* Poepp. et Endl., and *Nothofagus dombeyi* (Mirb.) Oerst. (Nothofagaceae) [2]. The female oviposits on the bark, the eggs hatch in about a month, and their larvae develop in two years, boring extensive galleries in the trunks without killing the tree but severely affecting the quality of the valuable wood in commercial farms [4,5,6,7]. The last instar larva builds a pupal chamber [3] connected to the outside by an orifice plugged with wood chips, which the adult later removes upon emergence. The pupae are found between July and December, and adults emerge between November and February [8]. In Valdivia Province (Los Ríos Region), infestation (% of attacked trees) in *N. obliqua* ranged from 14% to ca. 44%; in the provinces of Osorno and Llanquihue (Los Lagos Region), it reached almost 10%, while in La Araucanía Region, it ranged between 7% and 29%. The damage can range between 40 and 50% of the total volume of the log. Consequently, the attacked wood can only be used for wood chips or as fuel. Because of that, currently, *P. chilensis* is considered one of the most important pests in renewal (secondary) *N. obliqua* forests [9,10,11].

The control of these xylophagous as juveniles is usually not feasible using conventional insecticides, biological control agents, or cultural or physical measures, and no sanitary management measures are currently available against *P. chilensis* [12,13]. On the other hand, in the last decade, the ethological control of adults has been studied for the management of Cerambycidae in the northern hemisphere, with promissory reports of mass trapping and mating disruption. These specific, innocuous, and effective control techniques are based on the use of intraspecific attractant compounds produced by the insects [14,15] and include aggregation pheromones (emitted by males to attract both sexes, frequently occurring in most Cerambycidae) or sex pheromones produced by the female to attract only the male (uncommon in cerambycids) [16,17,18]. In Cerambycidae, long-range sex pheromones have been described for the orientation of conspecifics from relatively long distances to the emitting source. Short-range pheromones have also been reported, usually long-chain alkane compounds [19,20], allowing both the meeting of sexes and the specific recognition required to achieve successful mating [21].

Insects produce and emit volatile pheromones into the environment during the calling phase (i.e., when releasing long range chemical cues), the first behavior conducted in male–female encounters for mating purposes [21]. Calling occurs under specific environmental (e.g., temperature) and endogenous (e.g., age) conditions, which can be determined and then reproduced in the laboratory to collect pheromones from individuals maintained in controlled environments and identified with various analytical techniques [19]. Calling has been studied in males of some cerambycids [22,23] but very rarely in females [24,25]. After calling, during the approach and the courtship for specific recognition [21], short-range pheromones emitted by females play a role in several cerambycids [20]. Regarding *P. chilensis*, however, the sexual behaviors and chemical signals used to achieve effective encounters and mating are entirely unknown. Therefore, the objectives of our study were (a) to study the emergence and sexual ratio of adults of *P. chilensis* in the field, (b) to characterize the conditions during the approaches between both sexes for courtship/mating purposes, and (c) to capture and identify the respective volatiles emitted by virgin females and compounds of their terminalia, which will allow for laying the groundwork for technological developments for *P. chilensis* management.

## 2. Methodology

### 2.1. Localities, Study Seasons and Insect ID

Field activities were carried out during the spring–summer seasons between 2019 and 2023 at sites with *Proholopterus chilensis*-infested *N. obliqua* trees in Maquehue (−38.8364, −72.6941) (La Araucanía Region), at a “Universidad de La Frontera” (UF) experimental station located in the central valley, and at a private property at Llifén (−40.1978, −72.2598) (Los Ríos Region), a colder locality located 200 km further south in the foothills of the Andes. In addition, during the COVID-19 pandemic in 2019–2020, observations of the behaviors of *P. chilensis* males and females were conducted in Santiago (−33.447487, −70.673676), Metropolitan Region. The collection of volatiles, extraction of terminalia components, and identification of chemical compounds from females were carried out at the Chemical Ecology Laboratory of the UF in Temuco (−38.73965 −72.59842), La Araucanía Region. Insect identification (ID) was performed by the entomologist authors using only external morphological characters and confirmed by comparison with previously identified material at the MEUCh collection, Entomological Museum, the University of Chile, where two vouchers (males) were deposited. The determination of sexes considered the following features: female antennae shorter than the specimen’s body length, with antennae being longer in males; females with abdomens at least 25% wider than their elytras, as they are narrower or the same width in males; and males with two small spines merging from the margins of the last tergite, which is absent in females.

### 2.2. Emergence of P. chilensis from Trunks, Their Sex Ratio, and the Determination of the Attractive Sex and Attractiveness According to Age

To collect wild individuals from live *N. obliqua* trees, we fixed a prism (=cage, Figure 1) of ~10 × 20 cm made of wire (0.15 mm diameter) mesh (1.5 × 1.5 mm weft) over *P. chilensis* emergence orifices (~20 mm in greatest diameter) detected on the trunk (Figure 1). Approximately 800 cages (1–14 per tree) were installed between 2019 and 2023 (November to January each year) in Maquehue and Llifén, placed from ~0.4 to ~2 m in height. The cages were checked twice weekly (Maquehue) or daily (Llifén). When the cages contained specimens, these were taken to the laboratory and kept isolated in their respective cages at room temperature, with a cotton wick soaked in a 5% sugar solution used to feed the adults [25]. In this way, virgin *P. chilensis* of known age were sexed (as described by [3,7]) and kept for further use in indoor and field tests.

Virgin females or males of *P. chilensis* (aged 0–7 days from emergence) were individually placed in cages, as described above (one specimen/cage, one cage/trap). Each cage was stapled in the center of an interception trap (Alpha Scent, CA, USA, Figure 2). Traps have been successfully used in previous studies for adult cerambycid collections [26] and were hung in the field on live *N. obliqua* trees at a height of ~2 m, separated by ~20 m from each other, during the adult flight period (December to January). Three treatments were contrasted synchronously: female, male, and a control (trap without insects). After seven days in the field, the cumulative catches of *P. chilensis* were counted and the individuals sexed. A completely randomized block design (n = 8 replicates/treatment) was considered over three seasons between 2019 and 2023 (during the 2020–2021 season, no field work was possible because of the COVID-19 mandatory lock out). Given the structure of these results, no statistical analysis was conducted because in 2 out of 3 treatments catches were zero for all replicates, not providing variance for contrast purposes.

To determine the effect of a female’s age on her ability to attract males, we tested three treatments synchronously: individuals from 0 to 4, 5 to 9, or 10 to 14 days post-emergence from the trunk. Each replicate consisted of a trap arranged as described above, containing one female, which was kept for three days in the field during the peak of adult activity (January/2023). The cumulative captured congeners in three days were then counted and sexed. A completely randomized block design (n = 3) was considered, and the catches, trap, and day (CTD) were contrasted with the Kruskal–Wallis non-parametric and Tukey tests for mean comparisons (*p* = 0.05) [27].

### 2.3. Successful Behavioral Observations, Environmental Conditions, and Hours during P. chilensis Female–Male Sex Approaches

In the 2019–2020 season, at Santiago, transparent plastic cylinders ~70 cm long × ~13 cm section were used as indoor observation arenas. On the bottom, a mesh cotton strip was placed to allow the beetles to move and avoid the stress that they showed when trying to walk on the bare surface, as determined by preliminary observations. Another piece of mesh was arranged as a barrier blocking the section in the center of the cylinder, which allowed airflow (~0.5 m/s in the middle) generated by a fan located at one end, but not physical contact between the female (n = 24), placed upwind of the barrier) and the male (downwind). In addition, from 2021 to 2023, virgin females from 0 to 4 days old were individually tested in the field (Llifén) using (a) cages placed on interception traps (n = 12), as described in the previous section, and (b) free (not confined) specimens placed directly on the base of the trunk (n = 10). Only data for when males flew and actively approached the active (moving upward and downward and grooming themselves) females were considered for the analysis. Measurements included time (measured every 30 min), temperature, relative humidity (Cima control data logger OM-EL-USB-2, Omega brand, Deckenpfronn, Germany), light intensity (VWR International luxmeter, Radnor, PA, USA), and wind speed (EXTECH Instrument anemometer, Knoxville, TN, USA). A flashlight with a red filter was used during night observations (19:00 and 6:00 AM), allowing for the registering of the activity of males approaching females. No observations on windy or rainy nights were conducted in Llifén.

### 2.4. Capture of Volatiles and Collection of Compounds from the P. chilensis Terminalia of Females

Four head space dynamic (HSD) runs were made on *P. chilensis* females (n = 8) in the laboratory for five hours, i.e., between 23:00 and 4:00 AM, in darkness, under temperature and RH favorable for the approaching/encountering of both sexes (see results below). The HSD runs were conducted on individual females (n = 2) or groups of 2 or 4 (n = 1 ea.) specimens. They were confined to a 1325 mL Pyrex glass chamber, together with a fresh bark base obtained from live *N. obliqua* trunks, to facilitate insect displacement and avoid stress observed preliminarily in bare flasks, while the respective controls included either only bark or a male and bark (n = 4 in both). The capture of low-molecular-weight compounds was performed using a positive/negative pressure system inside the glass chamber, as described by [28], by pumping previously purified air over the insect and then draining it with another suction pump (flow rate of 1 L/min in both) into a Porapak Q column Water Corporation, Milford, MA, USA (100 mg; 80 Å 100 mesh). The compounds were then desorbed from the Porapak with hexane (GC grade; Merck, Darmstadt, Germany) (2 mL) or dichloromethane (GC grade; Merck, Darmstadt, Germany) (2 mL), obtaining two different extracts (not mixed). Before gas chromatography–mass spectrometry analysis (GC-MS), each extract was independently concentrated under a flow of nitrogen gas (70 mL/min) to 100 µL and analyzed [28,29]. 

The terminalia [30] exposed by four females (not observed in males) during the putative calling were excised with dissecting scissors and immediately fixed in a vial with hexane (GC grade). Then, before the filtration of the hexane containing the terminalia, Na_2_SO_4_ was added to remove water up to an extracted volume of 2000 µL and kept in an amber vial at −4 °C until GC-MS. In this study, the solvent alone was used as the control.

### 2.5. Identification of Volatiles and Terminalia Compounds from P. chilensis Females

The identification of compounds obtained from aerations and terminalia extracts was performed by GC-MS using a Thermo Electron Model Trace 1300 coupled to an ISQ 7000 Thermo Electron quadrupole mass spectrometer, with an integrated Xcalibur 4.1 data system (Thermo Fisher Scientific Inc., Waltham, MA, USA). In total, 1 µL of each extract was injected. An RTx-5MS column (30 m × 0.25 mm × 0.25 μm, Restek Corporation, Bellefonte, PA, USA) with helium as a carrier gas (1.0 mL/min) was used. The GC-MS procedure was performed with an initial temperature of 40 °C, which was maintained for 2 min and then increased at a rate of 5 °C/minute until 250 °C. The injector, transfer line, and ion trap temperatures were maintained at 250 °C. The mass detector used ionization energy of 70 eV. The mass range was from 30 to 500 amu. Compounds were identified by comparison with those in the NIST database ver. 2.3 (National Institute of Standards and Technology, Gaithersburg, MD, USA), and their retention indices were compared with those reported in the literature. Retention indices were calculated using an alkane series standard, as described by [31]. When identification by library could not be carried out, identification by mass/charge analysis was performed. A series of alkanes ranging from C_8_ to C_18_ was used and then extrapolated based on the seriality of the alkanes. The formula used was the following:$$K=100\left(Z-1\right)+\frac{LogRt\left(X\right)-LogRt\left(Z-1\right)}{LogRt\left(Z+1\right)-LogRt(Z-1)}$$
where Rt is the retention time, X is the unknown compound, Z − 1 is the alkane preceding the unknown compound, and Z + 1 is the alkane following the unknown compound.

## 3. Results

### 3.1. Emergence of Virgin P. chilensis from Trunks, Their Sex Ratio, and the Determination of the Attractive Sex and Attractiveness According to Age

During the four study seasons (2019–2020, 2021–2022, 2022–2023, and 2023–2024), 262 adults of *P. chilensis* emerging from trunks were collected from late November to late December (Maquehue) and from mid-December to mid-January (Llifén). A total of 79 females and 183 males were obtained, with a 1:~2.3 sex ratio. In interception traps baited with *P. chilensis* virgin females, 184 conspecific adult males were captured in total over the course of the study (Table 1), without capturing a single female. In contrast, in traps baited with live males and control traps, there were no captures of this species.

Females of *P. chilensis* attracted males throughout the tested age range, but significantly fewer (*p* = 0.036) for individuals older than nine days (2.7 ± 0.6 CTD of males) compared to those of 5–9 days of age (8.3 ± 1.5 CTD), suggesting an optimum of attraction in young adult specimens, since females of 0–4 days of age captured 6.7 ± 1.5 CTD, with no differences compared with the other two age categories. For this reason, we used individuals of less than ten days of age in subsequent studies. 

### 3.2. Successful Behavioral Observations, Environmental Conditions, and Hours during P. chilensis Female–Male Sex Approaches

All females placed freely on trunks, in cages, or in the plastic cylinders developed characteristic behaviors, including the projection of the terminalia, not observed among males. While stationary, females placed on trunks projected the terminalia and dragged it on the bark. After a few seconds doing so, the *P. chilensis* females retracted the terminalia and walked on the substrate for short distances (cm), stopping and projecting the terminalia again. This sequence was repeated many times by the females while climbing up and down the trunk. These behaviors were only observed during the period when males were actively flying (or walking, n = 24, in cylinders) during night hours, sometimes even around the box where the cages with females were taken to the field to set trials. Given the nocturnal sexual activity, only one approaching male was observed in detail when flying toward the female, showing casting flight, as described in [25], but males were observed when flying to a light source; in this case, the male flew in a straight line. Arriving males (n = 11) landed on the trunk and walked following the same route apparently marked by the female with her terminalia, whereas they (n = 10) were apparently disoriented after landing near the cages. Oriented approaches from males to putative calling females of *P. chilensis* occurred at 0–1 m/s (range of wind speed during observations), 30–50% RH, 0.1–0.3 lux, and 14.5–25.5 °C and between 22:30 and 5:00 AM in 2019 in Santiago, as well as at 0–1 m/s, 60–86% RH, 0–4 lux, and 11.6–21.3 °C and between 23:00 and 5: 00 AM in the period 2021–2023 in Llifén (Figure 3: pooled data obtained during observations of females placed freely on trunks, in cages, and in plastic cylinders).

### 3.3. Identification of Volatiles and Terminalia Compounds from P. chilensis Females

The analysis of chromatograms obtained from aeration extracts showed that the following compounds (Table 2) were present in females and absent in the respective controls: two oxygenated sesquiterpenes, a nitrogenous compound (C_20_), and a long-chain hydrocarbon (C_26_), plus an unidentified compound, all present at a 24.1 min retention time. In one female aerated individually, all five compounds were detected, and only three were found in the other one (those eluted at 24.1, 26.57, and 41.21 min); in the group of four females, only two compounds were found (at 24.1 and 26.57 min), and none were detected in the other group (a pair).

The fragments that explain this diagnosis for the family compound with a retention time of 24.87, i.e., an oxygenated sesquiterpene, were C_4_H_9_ for the fragment 57 *m*/*z*, and C_10_H_13_O_2_ for the fragment 165 *m*/*z*. For the family compound with a retention time of 26.57, the representative fragments were C_5_H_9_O for the 85 *m*/*z* fragment, C_5_H_9_ for the fragment 69 *m*/*z*, C_6_H_11_ for the fragment 83 *m*/*z*, and C_4_H_9_ for the fragment 57 *m*/*z*. In the case of the nitrogenated compound with a retention time of 41.21, the representative fragments correspond to C_15_H_16_N for the fragment 210 *m*/*z* and C_20_H_27_N for the fragment 281 *m*/*z*. We were not able to identify a more precise result because the parameters for a specific ID were too low.

An analysis of the extracts obtained from the terminalias of virgin *P. chilensis* females is presented in Table 3. The molecular weights, formulas, and retention indices of the compounds found are reported as fitting parameters > 90%. We detected high-molecular-weight linear alkanes, representing 67.48% (relative areas), and long-chain esters (29.6%). This identification was based on the characteristic fragmentation patterns of both classes of compounds but was not achieved for the compound whose retention time was 44.21 min.

## 4. Discussion

The collection of *P. chilensis* adults from trunks occurred in periods shorter than those reported in the literature [6,8]. The sex ratio found (1:2.3) among them is uncommon in Cerambycidae breeding, but no previous records exist for *P. chilensis*. Studies of other Cerambycidae species [32,33] mostly show parity in sexes (about 1:1) or sex ratios favorable to females. However, there are some reports of male-favorable sex ratios (1:2.2) obtained in cerambicinaes collected on flowers or at feeding or mating sites [34] and somewhat lower (1:1.5) ratios in species characterized by showing aggressive male competition for females, performing pre-mating walks (something observed in *P. chilensis*), or post-mating guarding [35]—behavior described by [36], as well as in another Chilean Cerambycidae species. We cannot rule out the possibility that some sampling effects affected our sex ratio results, but at least no sampling bias was expected regarding emergence hole size (similar between sexes) and the hole position along the trunk (random) used as criteria for installing cages. Moreover, cage placement started long before the onset and until the end of the flight and presence of adults in the field, discharging some possible temporal bias for emergence. Among the emerged adults, all were alive, and no mortality attributed to malformation, diseases or arthropod parasitoids, or predators was observed that might affect one sex more than the other one.

The fact that only males were captured in traps baited with virgin females, as well as the absence of *P. chilensis* captures in male-baited and control traps, preliminarily suggests that the females emit possible “volatile or airborne” pheromones to attract only males, a characteristic reported to date only in the subfamilies Prioninae and Lepturinae [20,30], but practically unknown in the subfamily Cerambycinae [18]. It is, however, important to point out that the taxonomic status of *P. chilensis* is uncertain: it has been classified as a Cerambycine [37], a Lepturinae [38], a Cerambycinae [1], and an *incertae sedis* [39] and mentioned as a case of misplaced taxa [40]. Nevertheless, in the last decade, there have been reports suggesting a deviation from this pattern in a few cerambicinaes. For instance, a compound collected in aerations of *Xylotrechus quadripes* (Chevrolat) females attracted only males, although it showed electroantennographic responses in both sexes [41]. Furthermore, a compound emitted by flowers of the orchid *Disa forficaria* Bolus attracted only males of *Chorothyse hessei* Quentin and Villiers that pollinate this plant [42], and two studies of cerambycid pest species in Asia, *Aromia bungii* Faldermann [43] and *Rhytidodera simulans* White [44], with extracts in the laboratory, did the same, suggesting a sex pheromone produced by the female. Thus, to the best of our knowledge, our preliminary findings in *P. chilensis* correspond to one of the first studies in Cerambycinae in which behavioral evidence (like also proposed for other species [44,45]) preliminarily suggests the existence of female-produced “volatile or airborne” pheromones, eventually acting as long-range sex pheromones (Table 2).

The mechanisms that could explain the *P. chilensis* female age effect on attractiveness to males are not known. However, a study of a Japanese cerambycid suggests that sexual maturity is reached in females after five days of adult life [46], but it should be considered that the *P. chilensis* females were prevented from mating (most were confined), which may extend the calling period in individuals that remain virgins. Additionally, in most studies with methodologies equivalent to the one used in ours, females up to 15 days from emergence were used, a period in which most maintained attractive capacity [45,47]. The literature points to calling behaviors occurring that start at one week of life in some Cerambycidae females: in *Anoplophora glabripennis* (Motschulsky) (Lamiinae), age did not affect calling in specimens aged 7–34 days old [48], and this behavior occurred between 7 and 24 days in *Glycobius speciosus* (Say) (Cerambycinae) [49].

Behaviors like those observed in *P. chilensis* females during the putative calling (terminalia projection and contact with the bark) have been reported in Cerambycidae [50,51].

The ultimate causes of these behaviors are unknown, but the fact that at least one male was observed approaching a (putative) calling female by casting flight in the field [zigzagging in 25] suggests responses to airborne semiochemicals, eventually including responses to female volatiles, host cues, or both. These responses did not occur below 11.6 °C, a factor that has also been reported to be restrictive in other cerambycids [52]. However, it is possible that the temperature during field tests did not exceed the minimum for either male-oriented flight (or walking) and/or for the performance of calling in females, so we cannot conclude that this reached either threshold. Methodologically similar studies [25], with males and females of a diurnal Cerambycidae species, also established a similar range of hours per day (5 h, with females prevented from mating during the hours of study).

Regarding the C_20_ and the C_26_ unspecifically identified compounds, these do not seem to be very promising as long-range chemicals since their molecular weight suggests low volatility (especially C_26_), making sesquiterpenes better candidates. With regard to the terpene compounds collected by HSD and identified generically, it should be noted that these types of chemicals have been reported as one of the main structural types of the pheromones present in cerambycid species [20] or as minor components of their pheromonal blends [53,54], but unlike our results, all were emitted by males. For instance, among monoterpenes, in *Megacyllene caryae* (Gahan), (S)-(−)-limonene has been identified as a component of the male-produced odor blend [55]. In *Megacyllene antennata* (White), (S)-α-terpineol, (S)-limonene, and terpinolene have been reported as well [56]. On the other hand, sesquiterpene hydrocarbons such as β-elemene, β-caryophyllene, α-humulene, and α-farnesene have been found on the surfaces of male and female elytras acting as contact pheromones, being them attractive to both sexes [57]. Furthermore, sesquiterpenes as fuscumol acetate ((E)-6,10-Dimethyl-5,9-undecadien-2-yl acetate; vapor Pressure: 1.61 × 10^−2^ mm Hg at 25 °C) have been reported to be attractants produced by males of many cerambycid species [58], mainly in Cerambycinae, and this particular compound was significantly attractive and identified by our group as probable components of a long-range aggregation pheromone for the South American cerambycid *Eryphus laetus* (Bl.) (Cerambycinae) [26]. The same article notes that under field conditions, this compound and two other sesquiterpenes (fuscumol and geranyl acetate) tested in the field individually did not attract *P. chilensis* despite conducting the experiment in a heavily infested site during their flying time, suggesting that these chemicals have no biological activity on this species. There are also some studies of nitrogen-containing semiochemicals (e.g., amides) for some compounds identified in arthropods [59,60] and at least one report of long-range pheromones produced by the female in the cerambicinae *Vesperus xatarti* Mulsant [61]. Finally, several C_26_ hydrocarbons (e.g., hexacosene) have been reported as cuticular pheromonal compounds (i.e., with contact but not long-range activity), but unlike in our study, they were produced by males of the species *Pidonia* spp. (Lepturinae) [62]. This is the first report of possible volatile or airborne chemicals and possible contact compounds obtained from *P. chilensis* females by HSD. It is now necessary to advance with identifying and evaluating these compounds in terms of their behavioral role and, eventually, as potential long- or short-range attractants.

Among the chemicals found in the terminalia, the following are considered relatively more volatile [63,64,65]: heneicosyl acetate, also reported in the profile of pods of *Vigna radiata* (L.) R. Wilczek (Fabaceae) and in the pre-anal glands of the lizard *Hemidactylus flaviviridis* Ruppell (Chordata: Squamata: Gekkonidae), and heptacosyl acetate, reported as an indicator compound for nematode infestation (*Anguina funesta* Price, Fisher and Kerr, Tylenchida: Anguinidae) in Poaceae [66]. As far as we know, no insect-associated pheromonal activity has been reported for these two chemicals [58]. On the other hand, docosyl acetate has been identified in labial glands of male bumblebees of the genus *Bombus* (Hymenoptera: Apidae), from which the sex pheromone is emitted [67], and has been listed as a minor component of the sex pheromone of the lepidopteran *Myelois cribrella* Hubner (Pyralidae), without electroantennographic activity [68]. The other compounds identified from terminalias have lower volatility: hentriacontane (PV: 1.40 × 10^−11^ mm Hg at 25 °C) has been reported in low proportions in the cuticles of adults of both sexes of *Cerambyx wellensi* Kuster (Cerambycidae: Lamiinae) [69], whose mixture of components evidenced roles in sexual behavior, but at a short range; heptacosane (PV: 2.81 × 10^−7^ mm Hg at 25 °C) has been detected in the elytra and thorax of females of two species of *Tetropium* (Spondylinidae) [70] and is recognized as a main pheromonal (contact) chemical, as it stimulates mating. Compounds of this type, with short-range activity, could serve as marking pheromones (trail-pheromones: widely described for ants and termites [71]) to guide males of *P. chilensis* walking on the substrate presumably previously marked by the female during calls, as observed in our field evaluations. This is like the behavior described in *Nadezhdiella cantori* (Hope) (Cerambycinae, [72]). Mixtures of long-chain alkanes (similar to those reported in Table 3) have also been described with short-range activity in females of several Cerambycidae subfamilies [20,51], which are detected by the antennae, maxillary palps, or tarsi of males in contact with female bodies or the marked substrate, something observed in males during our field tests. Together with long-range pheromones, these types of compounds complement a chemical communication system that enables specific encounters and recognition in some cerambycids [20,51]. These compounds might function during courtship, but no reports were found for Cerambycidae. Our study constitutes the first identification of compounds from the terminalia of *P. chilensis,* some of which could be part of the biosynthetic pathway of final compounds, and others could be degradation products [20] or have direct roles in the respective sexual behaviors and male orientation [51], which should be tested and elucidated in future research. All these compounds should be evaluated for their potential utility for *P. chilensis* management as well. A similar complex mate-finding system has been reported in cerambycids for *A. glabripennis* by [50,73], where males first attract females using a long-range pheromone, and then the male follows a trail-pheromone released by the female on the substrate. Interestingly, the trail marked with the latter compound is avoided by conspecific females and might act as a spacing pheromone to avoid intraspecific competition [50]. The putative trail-pheromone compound(s) produced by *P. chilensis* would help the arriving male to follow the female that first walks up and then down on the trunk. According to our findings, it is possible that chemical communication in the male–female approach/encounter in *P. chilensis* could be a hybrid. No reports like that were found for cerambycids, but a similar system has been described in some *Callosobruchus* beetles (Chrysomelidae) where females produce both types of pheromones [74], involving, first, a long- (as informed in compounds similar to those presented in Table 2) and then a short-range pheromone serving for recognition and, eventually, for courtship. In addition, both types of compounds are produced by the *P. chilensis* female, a mechanism that needs to be confirmed and, as far as we know, is not reported in the literature for cerambycids.

## 5. Conclusions

*Proholopterus chilensis* that emerged from live *Nothofagus obliqua* trunks showed a sex ratio highly favorable to males, suggesting intense competition for females.

Only females of *P. chilensis* are attractive to males, which were not attractive to either sex, something unusual in Cerambycinae.

Females of *P. chilensis* attract males during the night, a period that can extend for up to 5 h in the absence of mating, and their attractiveness depends, at least, on their age and environmental temperature.

The analysis of extracts from aerations of *P. chilensis* females allowed the generic identification of four volatile compounds, while five compounds were specifically identified from their terminalias. These compounds formed a complex and complementary chemical communication system in which the first ones would attract males from relatively long distances. Then, the compounds found in the terminalia helped the female mark the trail (on the trunk) towards her location during the putative calling/courtship phase so that the encounter with males could occur. This hypothesis should be further tested, as should the compounds found for their possible role in sexual behavior and as potential attractants for *P. chilensis* management.

## Figures and Tables

**Figure 1 insects-15-00741-f001:**
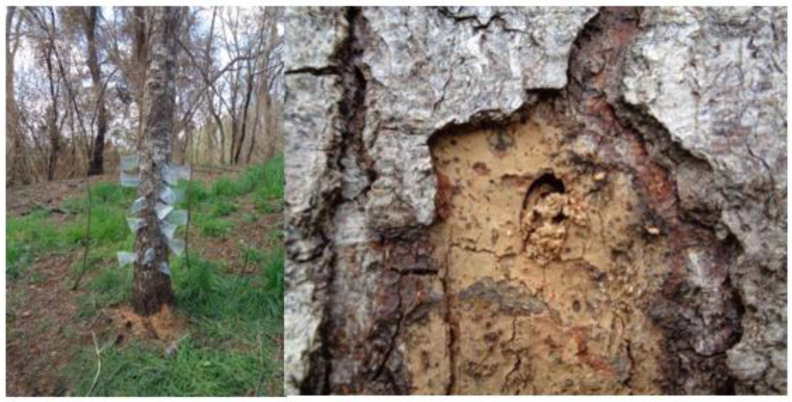
Wire mesh cages (n = 14) placed on a *N. obliqua* tree for capturing virgin *P. chilensis* adults during emergence. Sawdust produced by the larval activity can be seen accumulated at the base of the trunk (**left**); a *P. chilensis* emergence hole blocked with wood chips (**right**).

**Figure 2 insects-15-00741-f002:**
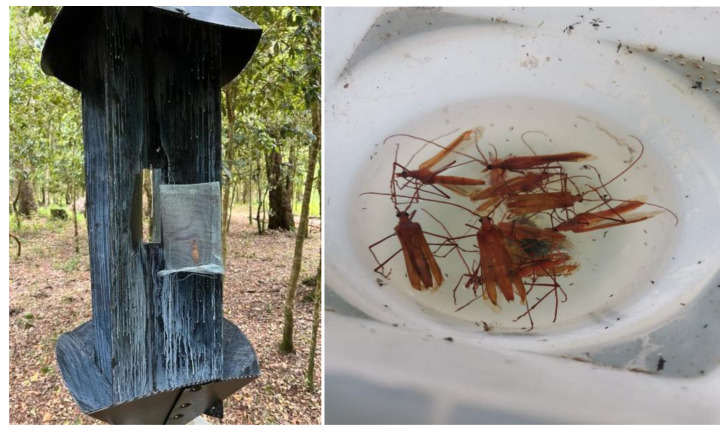
A *Proholopterus chilensis* female placed in a cage stapled to a flight interception trap (**left**); males captured in the trap’s collecting cup (**right**).

**Figure 3 insects-15-00741-f003:**
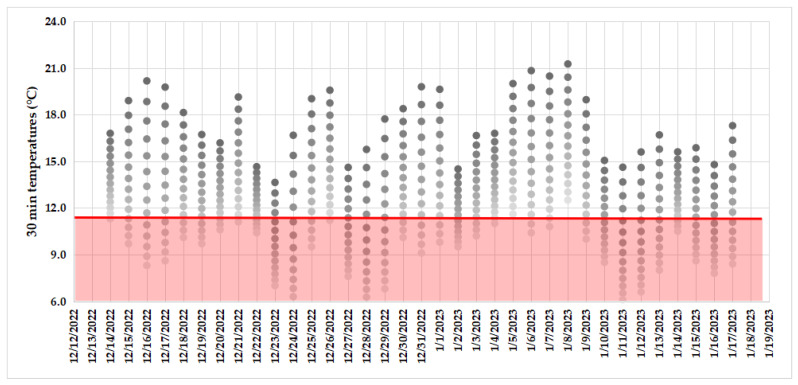
Pooled data (measured in Santiago—females in cylinders—and Llifén—field trials with females in cages and free on the trunk) on temperatures during the oriented approaches of males to females of *P. chilensis* taken every 30 min (dots in each column inside the graph) for each night of observation. The grey tone of the dot varies from light to dark as the night hour progresses. The pink area delimited by the red line represents the temperatures in which no male–female approaches were observed.

**Table 1 insects-15-00741-t001:** Male *Proholopterus chilensis* specimens captured in “traps per treatment” (n/season) during the 2019–2023 period.

	2019–2020 n = 4	2021–2022 n = 3	2022–2023 n = 4	Total Captures
Control	0	0	0	0
Male	0	0	0	0
Female	58	80	46	184

**Table 2 insects-15-00741-t002:** Analytical features and compounds identified at the generic level by GC-MS in aeration extracts obtained by HSD from virgin females of *Proholopterus chilensis*.

Retention Time (min)	Family Compound Name ^1^	Area ^2^ (%)	Main Mass Fragments (*m*/*z*)
24.1	Not identified	6.23	57 (100), 93 (76), 81 (70), 96 (63), 73 (56), 95 (53), 107 (48), 79 (46), 41 (46), 91 (40)
24.87	Oxigenated sesquiterpene 1	53.41	57 (100), 161 (79), 71 (59), 45 (54), 41 (39), 203 (37), 85 (36), 133 (32), 77 (31)
26.57	Oxigenated sesquiterpene 2	4.30	57 (100), 69 (89), 205 (83), 71 (57), 83 (40), 111 (37), 85 (100), 70 (35), 41 (30)
41.21	Nitrogenated compound (C_20_)	33.17	210 (100), 211 (17), 281 (94)
43.77	Long-chain hydrocarbon (C_26_)	2.88	43 (100), 57 (700), 71 (680), 85 (600), 41 (418), 71 (144)

^1^: Due to the fragmentation pattern, only the chemical family can be suggested. ^2^: The peak chromatogram relative area.

**Table 3 insects-15-00741-t003:** Analytical features and compounds identified by GC-MS in terminalia extracts obtained from virgin females of *Proholopterus chilensis*.

Retention Time (min)	Compound Name	Area ^1^ (%)	MW ^2^ (g/mol)	Chemical Composition	RI ^3^	Main Mass Fragments (*m*/*z*)
40.03	Heneicosyl acetate	14.12	354	C_23_H_46_O_2_	2515	43 (100), 83 (88), 57 (86), 97 (81), 55 (10), 69 (62), 71 (54), 61 (49), 41 (41), 70 (39)
43.29	Docosyl acetate	8.78	368	C_24_H_48_O_2_	2618	43 (100), 57 (64), 97 (59), 83 (58), 55 (57), 61 (50), 69 (46), 111 (34), 41 (32), 71 (31)
44.21	Not identified	2.90	-	-	-	130 (100), 117 (59), 131 (54), 95 (36), 55 (30), 57 (29), 43 (28), 81 (27), 71 (26)
44.82	Heptacosane	22.69	380	C_27_H_56_	2700	57 (100), 71 (85), 85 (63), 43 (50), 99 (26), 55 (24), 41 (19), 69 (19), 113 (16), 83 (15)
48.21	Heptacosyl acetate	6.71	438	C_29_H_58_O_2_	3071	97 (100), 57 (94), 83 (91), 69 (77), 55 (75), 43 (61), 71 (59), 111 (54), 41 (40), 85 (39)
48.87	Hentriacontane	44.79	436	C_31_H_64_	3100	57 (100), 71 (86), 85 (64), 43 (48), 99 (27), 55 (23), 69 (18), 41 (17), 113 (15), 83 (13)

^1^: peak chromatogram relative area; ^2^: molecular weight; ^3^: retention index.

## Data Availability

The chromatograms obtained in this study are available on request from the corresponding author.

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
