# Peer review of "Factors to Male-Female Sex Approaches and the Identification of Volatiles and Compounds from the Terminalia of Proholopterus chilensis (Blanchard) (Coleoptera: Cerambycidae) Females in Nothofagus obliqua (Mirb.) Oerst. (Nothofagaceae) Forests in Chile"

_insects, 2024, doi:10.3390/insects15100741_

Round 1
Reviewer 1 Report
Comments and Suggestions for Authors
The manuscript corresponds to the first exploration of volatile emitted by females of Proholopterus chilensis as a potential sex pheromone blend. The research is quite interesting and descriptive but there are some important facts that should be considered.
Introduction
Even if you mentioned the infestation of this insect reduces wood quality for commercial purposes, please, extend an explanation of how important this pest. Why this pest was worth to be scrutinized? For instance, in terms of production-economic losses: The attacks of these insects make not worth to continue with wood obtention for producers? or they can continue obtaining enough income? The population density, attack frequency and damage is increasing through time?
Materials and methods
You should describe in more detail your trapping conditions.
Line 137. You mention n = 8. Does this mean 8 traps with virgin females? or 8 virgin females per traps? In the first case, then, how many females per trap? In the second case: How many replicates per treatment? I deduced one female per trap and replicated 8 times until I saw Figure 2A. So please, explain it directly in the proper section.
Line 137-138: "Given the structure of the results during... no statistical analysis was required in this case". What is the structure of your results? Why does this make unnecessary statistical analysis? If you replicated this for three years, at least you can look if insect catches increased, decreased or remained constant during that period. In fact, you do not show any table or graph showing catches variation during three evaluated years. That would be interesting to watch, or at least to be mentioned and described in the text.
In the last part of the same section you mentioned a comparison based on catches/trap/day, but how long was the trapping period in each season? one week, two week, one month or....? Did you count trapping daily or every week? Please, explain it clearly and directly in material and methods.
Lines 181-182: Were the hexane and dichloromethane extracts mixed after the cartridge elution and then concentrated? Was the mixture of them the analyzed samples by GC-MS? Please explain it.
Lines 173-176: When you mention you used blanks without insects, were those just empty flasks? or do you have any blank without insects but with tree bark in the flask? As you mentioned insects were posed over bark to avoid stress, then you can precisely discard that all detected volatiles come from the insects and not from the bark too.
Lines 203-204: Please, at least indicate what alkane series you used and that you used standard Kovats' formula for linear separation conditions, then, readers do not need to look another paper to get to know what you did.
Results
In general, I really liked that all behavior was described with detail. Nevertheless, In the manuscript you frequently described you found unusual facts, for instance, sex ratio or calling females, whihc is interesting. I also noticed that you mentioned several times "unpublished data". In my opinion, stating "unusual findings" and "unusual behavior" make them quite interesting and get the reader's attention. Nonetheless, if you only make a description in the text but you do not show any clear evidence, this makes them kind of uncomplete findings. Thus, as behavior is one of you most important results, I would like to see pictures or videos of your descriptions. For example, I think it is more important showing a picture of a calling female with and exposed gland than showing figure 3, which can be supplementary material. Then, replace those "unpublished data" at least with supplementary pictures or videos.
In materials in methods, you claimed to perform compound annotation by mass fragmentation interpretation. In results, you mentioned there were two oxygenated sesquiterpenes and a nitrogenated compound, but you do not explain what fragments are diagnostic to determine those type of compounds or any fragmentation mechanism specific for those compounds features, so please add it.
Discussion
In discussion, you mentioned C20 and C26 compounds as part of the long-range pheromone. However, as those compounds were not identified you do not know their vapor pressure, and according to their molecular weight there are not very volatile, especially C26. Thus, this compounds do not seem to be very promissory for long-range effects, where very volatile compounds are needed. I think this sesquiterpenes has more potential in this sense. Please consider this fact in the discussion with appropriate references.
Finally, you do not show any gas chromatogram of the volatile extracts, please add them at least as supplementary materials.
Lines 321-323: Please before going directly to sesquiterpenes, add a bit of discussion of the presence of terpenes (mono or sesqui) in the pheromone composition of Cerambycidae and other insects. Please, mention the name of the compounds found as part of the pheromone, discuss if structurally they are similar to yours (since sesquiterpenoids can show variety of functionalization). For instance, if those compounds are identified in those studies, you can check their fragmentation patterns in the NIST library and check if there are similarities between the and your unidentified sesquiterpenes, then state the possibility of new sesquiterpenoids produced by Proholopterus chilensis. I mean, you do not mention if your sesquiterpenoids did not show any match with the library or at least a match with low score.
Hexacosane is not a short-range pheromone compound but a direct contact pheromone compound since it is identified by direct physical contact, please use adequate terms.
Please add some discussion and references about the typical components of the tissue of sexual glands, since several of them are alkanes or fatty acid like compounds.
Author contributions
You mention some author names for insect ID, which I think means insect identification, but you do not mention in the material and methos section how insect identification was performed. Was this made by taxonomic characters or by molecular means? Also, please indicate the name of who performed the identification in the case of taxonomic identification. Also indicate if there is a boucher number and where the insect specimens were deposited. Finally, please indicate what character you used to distinguish between male and female specimens.
Comments on the Quality of English LanguageLittle eddition rchanges are needed
Author Response
Comment 1. Even if you mentioned the infestation of this insect reduces wood quality for commercial purposes, please, extend an explanation of how important this pest. Why this pest was worth to be scrutinized? For instance, in terms of production-economic losses: The attacks of these insects make not worth to continue with wood obtention for producers? or they can continue obtaining enough income? The population density, attack frequency and damage is increasing through time?
Response. Besides what´s already in the manuscript we provide the following paragraph to explain the species importance (in quotes and italic ). In lines 74-78.
"The damage can range between 40 and 50% of the total volume of the log. Consequently, the attacked wood can only be used for wood chips or as fuel. Because of that, currently P. chilensis is considered one of the most important pest in renewal (secondary) N. obliqua forests"
However, we did not find an economic analysis.
Comment 2. You should describe in more detail your trapping conditions.
Line 137. You mention n = 8. Does this mean 8 traps with virgin females? or 8 virgin females per traps? In the first case, then, how many females per trap? In the second case: How many replicates per treatment? I deduced one female per trap and replicated 8 times until I saw Figure 2A. So please, explain it directly in the proper section.
Response. The next sentence was added to the methodology. In line 149
"One specimen/cage, one cage/trap, 8 replicates/treatment"
Comment 3. Line 137-138: "Given the structure of the results during... no statistical analysis was required in this case". What is the structure of your results? Why does this make unnecessary statistical analysis?
Response. In italic the sentence added to the text in manuscript. Lines 155-157
"Given the structure of these results no statistical analysis was conducted because in 2 out of 3 treatments catches were zero in all replicates, not providing variance for contrast purposes."
If you replicated this for three years, at least you can look if insect catches increased, decreased or remained constant during that period. In fact, you do not show any table or graph showing catches variation during three evaluated years (added). That would be interesting to watch, or at least to be mentioned and described in the text. …. n<
Data is now provided in Table 1, Line 248
Table 1. Male Proholopterus chilensis captures in “traps per treatment” (n/season) during the 2019-2023 period.
|
2019-2020 |
2021-2022 |
2022-2023 |
Total captures |
Control |
0 |
0 |
0 |
0 |
Male |
0 |
0 |
0 |
0 |
Female |
58 |
80 |
46 |
184 |
Comment 4. In the last part of the same section you mentioned a comparison based on catches/trap/day, but how long was the trapping period in each season? one week, two week, one month or....? Did you count trapping daily or every week? Please, explain it clearly and directly in material and methods.
Response. We rephrased the sentence for clarity as follows Lines 155-157:
"After seven days in the field, the cumulative catches of P. chilensis were counted and individuals were sexed".
Comment 5. Lines 181-182: Were the hexane and dichloromethane extracts mixed after the cartridge elution and then concentrated? Was the mixture of them the analyzed samples by GC-MS? Please explain it.
Response. No mix was made. Every extract was processed and analyzed independently. The new sentence (in italic below) should clarify that. Lines 208-212:
"...... (2 mL) or dichloromethane (GC grade; Merck, Darmstadt, Germany) (2 mL), obtaining two different extracts (not mixed). Before gas chromatography-mass spectrometry analysis (GC-MS), each extract was independently concentrated under a flow of nitrogen gas (70 mL/min) to 100 µL and analyzed [28,29]."
Comment 6. Lines 173-176: When you mention you used blanks without insects, were those just empty flasks? No, we didn´t use empty flasks or do you have any blank without insects but with tree bark in the flask? YES.
Response. Lines 200-203
"... were confined in a 1,325 mL Pyrex glass chamber, always together with a bark base obtained from live N. obliqua trunks, to facilitate insect displacement and avoid stress observed preliminarily in HSD bare flasks. The controls were either only bark or an adult (a female or a male)+bark."
Comment 7. Lines 203-204: Please, at least indicate what alkane series you used and that you used standard Kovats' formula for linear separation conditions, then, readers do not need to look another paper to get to know what you did.
Response. Lines233-236
"A series of alkanes ranging from C8 to C18 was used, and then extrapolated based on the seriality of the alkanes. The formula used was: K = 100(Z-1) + {[(Log Rt(X)) - (Log Rt(Z-1)]/[(Log Rt(Z+1)-(Log Rt(Z-1)]}, where Rt is the retention time; X is the unknown compound; Z-1 is the alkane preceding the unknown compound; and Z+1 is the alkane following the unknown compound."
Comment 8. In general, I really liked that all behavior was described with detail. Nevertheless, In the manuscript you frequently described you found unusual facts, for instance, sex ratio or calling females, whihc is interesting. I also noticed that you mentioned several times "unpublished data"
Responses. Phrase (unpublished data) deleted from the manuscript in lines 240, 251, 392, 408
In my opinion, stating "unusual findings" and "unusual behavior" make them quite interesting and get the reader's attention. Nonetheless, if you only make a description in the text but you do not show any clear evidence, this makes them kind of uncomplete findings. Thus, as behavior is one of you most important results, I would like to see pictures or videos of your descriptions. For example, I think it is more important showing a picture of a calling female with and exposed gland than showing figure 3, which can be supplementary material. Then, replace those "unpublished data" at least with supplementary pictures or videos.
We have videos showing all behaviors. However, we are preparing another paper on behavioral responses of P. chilensis, so it is unpublished material. We did add a graphical abstract that should explain the behavioral sequence. Line 56
Comment 9. In materials in methods, you claimed to perform compound annotation by mass fragmentation interpretation. In results, you mentioned there were two oxygenated sesquiterpenes and a nitrogenated compound, but you do not explain what fragments are diagnostic to determine those type of compounds or any fragmentation mechanism specific for those compounds features, so please add it.
Response. This paragraph (see below) was added by the chemist coauthors to anwser this comment. Lines 300-307
"The fragments that explain this diagnosis for the compound with retention time 24.87, an oxygenated sesquiterpene, were: fragment C4H9 for 57 m/z, and C10H13O2 for the fragment 165 m/z. For the compound with retention time 26.57 the representative fragments were: C5H9O for the 85 m/z fragment, C5H9 for fragment 69 m/z, C6H11 for fragment 83 m/z, and C4H9 for fragment 57 m/z. In the case of the nitrogenated compound with retention time 41.21, the representative fragments correspond to C15H16N with 210 m/z and C20H27N for the fragment 281 m/z."
Comment 10. In discussion, you mentioned C20 and C26 compounds as part of the long-range pheromone. However, as those compounds were not identified you do not know their vapor pressure, and according to their molecular weight there are not very volatile, especially C26. Thus, this compounds do not seem to be very promissory for long-range effects, where very volatile compounds are needed. I think this sesquiterpenes has more potential in this sense. Please consider this fact in the discussion with appropriate references.
Response. The next sentences was added to address this comment, and placed at the beggining of the discussion of this part of the article. Lines 382-384
"Regarding the C20 and the C26 unspecifically identified compounds, these do not seem to be very promissory as long-range chemicals since their molecular weight suggests low volatility (especially C26), being then sesquiterpenes better candidates. Afterthat, there was already a discussion citing several references on sesquitepenes as cerambicid long-range pheromones."
Comment 11. Finally, you do not show any gas chromatogram of the volatile extracts, please add them at least as supplementary materials.
Responses. Selected chromatograms are provided in the separated file.
Comment 12. Lines 321-323: Please before going directly to sesquiterpenes, add a bit of discussion of the presence of terpenes (mono or sesqui) in the pheromone composition of Cerambycidae and other insects.
Done as follows below: Lines 388-394
"Among monoterpenes, in Megacyllene caryae (Gahan), (S)-(−)-limonene has been informed as a component of the male-produced odor blend [56]. In Megacyllene antennata (White), (S)-α-terpineol, (S)-limonene and terpinolene have been reported as well [57]. On the other hand, sesquiterpene hydrocarbons such as β-elemene, β-caryophyllene, α-humulene, and α-farnesene have been found on the surface of male and female elytra acting as contact pheromone, attractive to both sexes [58]."
Comment 13. Please, mention the name of the compounds found as part of the pheromone, discuss if structurally they are similar to yours (since sesquiterpenoids can show variety of functionalization) For instance, if those compounds are identified in those studies, you can check their fragmentation patterns in the NIST library and check if there are similarities between the and your unidentified sesquiterpenes, then state the possibility of new sesquiterpenoids produced by Proholopterus chilensis. I mean, you do not mention if your sesquiterpenoids did not show any match with the library or at least a match with low score.
Response. We did it, but the probability for specific ID was too low (2-4%) and other parameters as the match (434) or reverse match (494) were way below 800.
Comment 14. Hexacosane is not a short-range pheromone compound but a direct contact pheromone compound since it is identified by direct physical contact, please use adequate terms
Response. The sentence was rewritten as follows (line in bolds in the original text, see below). Line 408
Finally, several C26 hydrocarbons (e.g. hexacosene) have been reported as cuticular pheromonal compounds (i.e. with contact, not long-range activity), but unlike our study, produced by males in the species of Pidonia spp (Lepturinae) [62].
Comment 15. Please add some discussion and references about the typical components of the tissue of sexual glands, since several of them are alkanes or fatty acid like compounds.
Response. We did a literature search in Google Academic using the next combination of key words: cerambycidae terminalia tissue extracts, cerambycidae sex-gland extracts, cerambycidae contact pheromone elytra, cerambycidae contact pheromone thorax, cerambycidae contact pheromone abdomen, up to the first 100 references, finding one reference [57] already cited before in the manuscript Lines 392-394.
Comment 16. You mention some author names for insect ID, which I think means insect identification, but you do not mention in the material and methos section how insect identification was performed. Was this made by taxonomic characters or by molecular means? Also, please indicate the name of who performed the identification in the case of taxonomic identification. Also indicate if there is a boucher number and where the insect specimens were deposited. Finally, please indicate what character you used to distinguish between male and female specimens
Response. In lines 123-130. Only external morphology was used for insect ID (identification) performed by the entomologists authors who have worked for more than 30 years with the chilean cerambicifauna. Besides, the specimens were compared with previously identified material at the MEUCh (Entomological Museum, University of Chile) reference collection where two males were deposited as vouchers. The morphological features used to sex specimens were: "female antennae shorter than its body length, being longer in males; females present abdomen at least 25% wider than elytra and it is narrower or the same width as elytra in males; males present two spurs merging from the margins of the last tergite absent in females".
Reviewer 2 Report
Comments and Suggestions for Authors
The work presented in the manuscript includes interesting observations and preliminary experiments concerning the chemical ecology of Proholopterus chilensis (Cerambycidae) a pest of Nothofagus obliqua. While there appears to be much data worthy of publishing in this manuscript, some improvements to the presentation and clarification of the results are needed.
Below are some comments of varying importance for improving the manuscript:
Ln137-138 “Given the structure of the results during 137 four seasons (2019-2023), no statistical analysis was required in this case.” Unclear why not? Is it just because of the overwhelming sex bias? There are still binomial or categorical statistical tests that could be performed and should be.
Section 3.1: Do you have any hypotheses for the biased sex ratio. Could there be biases in the sampling procedure, maybe if holes are not randomly selected for enclosure for some reason like the temporal and special distribution of the creation of these holes by sex varying? Is female survivorship lower, or would there be reasons for a biased sex ratio? I understand all of this would be speculative, but ruling out some of these possibilities would be helpful if you could.
Section 3.2: The total numbers of observations made are not presented. The paragraph mentions one flying approach observed. Was this out in the field or in the caged female experiments in the lab One possible anemotactic flight is interesting but is far from suggestive of a pheromone. It is possible that the males are using host cues, unrelated to the females.
The authors mention three observational experiments in section 2.3. There were lab experiments where females were presented to males in a wind tunnel bioassay. There were also field observations of females placed in cages near trees or freely roaming near trees. There is no indication of the sample sizes of each experiment? This is needed both in terms of the numbers of such assays run and the successful behavioral events observed.
The authors mentioned that they were always above the temperature threshold, but other info is not included. In addition to how many approaches, what is the average temperature reading when approaching? All of this data could and should be provided. Is Figure 3 the pooling of these three observational experiments?
Figure 3: Presentation is unclear. It seems there are multiple observations each night? The X axis should have the dates listed perpendicularly reading down from each point so that they can be visually matched with the specific data more easily. It would also be preferable to not have full gridlines, but rather major and minor tick marks (Major for the points where a label is provided, minor between)
Similarly for the Y axis the caption is unclear. Instead of “horary temperatures” wouldn’t “30 min temperature” be more accurate? The “per date” designation also is not needed on the Y axis as the X axis provides the date. Also, the shading does not always seem apparent in each day. Perhaps you could use a greater contrast from white to black if you had circles with lined borders when they are white?
Table1: The identification of oxygenated sesquiterpines is interesting, along with some other volatile compounds. How many replicates of the collection assay showed these compounds. How many control and male aerations were performed?
Because the response to an airborne pheromone was not definitively proven and the range of any potential chemical attraction is unknown, it would be better to refrain from using the terms “long-range pheromone” and “short range pheromone” in the discussion until more data is collected on this species. At this point, the data is allowing you to speculate on possible “volatile or airborne” pheromones vs likely “contact or trail’ pheromones. The length of the range of the potential pheromone may greatly affect its usefulness for control approaches, and thus this paper could become confusing to read once more is understood about the system after further research.
Comments on the Quality of English LanguageThere were no significant problems
Author Response
Comment 1. Ln137-138 “Given the structure of the results the seasons (2019-2023), no statistical analysis was required in this case.” Unclear why not? Is it just because of the overwhelming sex bias? There are still binomial or categorical statistical tests that could be performed and should be.
Response. The original explanation was modify (in quotes and italic) for clarity. Please Lines 159-161.
"Given the structure of these results no statistical analysis was conducted because in 2 out of 3 treatments catches were zero in all replicates, not providing variance for contrast purposes."
Comment 2. Do you have any hypotheses for the biased sex ratio. Could there be biases in the sampling procedure, maybe if holes are not randomly selected for enclosure for some reason like the temporal and special distribution of the creation of these holes by sex varying? Is female survivorship lower, or would there be reasons for a biased sex ratio? I understand all of this would be speculative, but ruling out some of these possibilities would be helpful if you could.
Response. In the original text, we presented some possible explanations, based on a literature review, for this unusual result. Following reviewer 2, we now added a paragraph (see below in quotes and italic) with some arguments to support our belief that there was no bias during both, the installation of cages or during the collection of specimens. Lines 328-335
"We cannot rule out some sampling effects affected our sex ratio results, but at least no sampling bias was expected regarding emergence hole size (similar between sexes) and the hole position along the trunk (random) used as a criteria for installing cages. Besides, cages placement started long before the onset and until the end of the flight and presence of adults in the field, discharging some possible temporal bias for emergence. Among the emerged adults, all were alive, and no mortality attributed to malformation, diseases or arthropod parasitoids or predators was observed that might affect one sex more than the other one."
Comments 3.The total numbers of observations made are not presented.
Responses. The number (n) per type of experiment were added in Line 182, 185 y 186.
The paragraph mentions one flying approach observed. Was this out in the field or in the caged female experiments in the lab.
This is now explained in Line 372
One possible anemotactic flight is interesting but is far from suggestive of a pheromone. It is possible that the males are using host cues, unrelated to the females (we added this possibility as well, text in bolds below).
The sentence was rewritten as follows in Line 373
"The ultimate causes for these maneuvers are unknown but the fact that at least one male was observed approaching a calling female by casting flight in the field [zigzagging in 25] suggests responses to airborne semiochemicals, eventually including responses to female volatiles, host cues or both."
Comment 4. The authors mention three observational experiments in section 2.3. There were lab experiments where females were presented to males in a wind tunnel bioassay. There were also field observations of females placed in cages near trees or freely roaming near trees. There is no indication of the sample sizes of each experiment?
Response. We added de “n value” in each case in methodology (Lines 182, 185, 186) and results (Lines 266, 271, 272), see below (in bolds).
............... but not physical contact between the female (n= 24), placed upwind of the barrier) and the male (downwind). In addition, from 2021 to 2023, virgin females from 0 to 4 days old were individually tested in the field (Llifén) using: a) cages placed on interception traps (n = 12), as described in the previous section, and b) free (not confined) specimens placed directly on the base of the trunk (n = 10).
These behaviors were only observed during the period males were actively flying (or walking, n = 24, in cylinders) during night hours, sometimes even around the box where the cages with females were taken to the field to set trials. Given the nocturnal sexual activity, only one approaching male was observed in detail when flying toward the female, showing casting flight as described in [25] but not observed on males when flying to a light source; in this case, the male flies in a straight line. Arriving males (n = 11) land on the trunk and walk following the same route apparently marked by the female with her terminalia, whereas they (n = 10) were apparently disoriented after landing near the cages.
Comment 5. The authors mentioned that they were always above the temperature threshold, but other info is not included (??). In addition to how many approaches, what is the average temperature reading when approaching? All of this data could and should be provided. Is Figure 3 the pooling of these three observational experiments?
Response. By doing these observations we were tying to indirectly establish the minimum temperature necessdary during for HSD runs, since (we assume) males only actively respond when females are calling. Thus, we can not assure 11,6 °C is either the male (for oriented flight or walking) or the female (for calling) trehshold. With respect to the second part of this comment, since temperatures were taken every 30 minutes, we cannot be precise on the exact temperature during each male approach; a rough estimation is that male responses mostly occurred between 15-18°C. Since it is a bit vague we decide not to include this comments in the new version. Regarding the last question for comment 5, YES, it is correct and explicit now with the figure label. Please see Line 182
Comment 6. Figure 3: Presentation is unclear. It seems there are multiple observations each night? The X axis should have the dates listed perpendicularly reading down from each point so that they can be visually matched with the specific data more easily. It would also be preferable to not have full gridlines, but rather major and minor tick marks (Major for the points where a label is provided, minor between (??). Similarly for the Y axis the caption is unclear. Instead of “horary temperatures” wouldn’t “30 min temperature” be more accurate? The “per date” designation also is not needed on the Y axis as the X axis provides the date. Also, the shading does not always seem apparent in each day. Perhaps you could use a greater contrast from white to black if you had circles with lined borders when they are white?
Response. Regarding the first question, it is correct. All the suggested improvements, as far as we were able to, were incorporated to both, the new figure (Line 280) and the new legend (Line 282) (see below) .
Figure 3. Pooled data (measured in Santiago – females in cylinders- and Llifén – field trials with females in cages and free on the trunk) on temperatures during oriented approaches of males to females of P. chilensis taken every 30 minutes (dots in each column inside the graph) each night of observation. The grey tone of the dot varies from light to dark as the night hour progresses. The pink area delimited by the red line represents the temperatures in which no male-female approaches were observed.
Comment 7. Table1: The identification of oxygenated sesquiterpines is interesting, along with some other volatile compounds. How many replicates of the collection assay showed these compounds . How many control and male aerations were performed.
Responses. The anwsers are highlighted with the text below.
Regarding the runs: see Methodology, Lines 199 y 203
"Four head space dynamic (HSD) runs were made on P. chilensis females in the laboratory for five hours, between 23:00 and 4:00 AM, in darkness, under temperature and RH favorable for the approach/encounter of both sexes (see results below). The HSD runs were conducted on individual females (n = 2) or groups of 2 or 4 (n = 1 ea.) specimens. They were confined in a 1,325 mL Pyrex glass chamber, together with a fresh bark base obtained from live N. obliqua trunks, to facilitate insect displacement and avoid stress observed preliminarily in bare flasks while the respective controls included either only fresh bark male+bark (n = 4 in both)."
Regarding the compounds/run, see Results, Lines 292-296
"In one female aerated individually, all five compounds were detected and only three were found in the other one (those eluted at 24.1, 26.57, and 41.21 min); in the group of 4 females, only two compounds were found (at 24.1 and 26.57 min) and none were detected in the other group ( a pair)."
Comment 8. Because the response to an airborne pheromone was not definitively proven and the range of any potential chemical attraction is unknown, it would be better to refrain from using the terms “long-range pheromone” and “short range pheromone” in the discussion until more data is collected on this species. At this point, the data is allowing you to speculate on possible “volatile or airborne” pheromones vs likely “contact or trail’ pheromones. The length of the range of the potential pheromone may greatly affect its usefulness for control approaches, and thus this paper could become confusing to read once more is understood about the system after further research.
Response. We agree with reviewer 2. Because of that we relativize those particular findings as "preliminary" along the text (Lines 179, 351). Besides, when appropriate we replaced the respective terms for either "possible volatile or airborne pheromones" (instead of long-range pheromone, Line 410) or "likely contact or trail pheromone" (instead of short-range pheromone, Line 48) as suggested.
Round 2
Reviewer 2 Report
Comments and Suggestions for Authors
The authors provided a thoughtful and thorough response to the reviewer comments